# Influence of Aging, Macronutrient Composition and Time-Restricted Feeding on the Fischer344 x Brown Norway Rat Gut Microbiota

**DOI:** 10.3390/nu14091758

**Published:** 2022-04-22

**Authors:** Abbi R. Hernandez, Keri M. Kemp, Sara N. Burke, Thomas W. Buford, Christy S. Carter

**Affiliations:** 1Division of Gerontology, Geriatrics and Palliative Care, Department of Medicine, University of Alabama at Birmingham, Birmingham, AL 35294, USA; twbuford@uabmc.edu (T.W.B.); cartercs@uabmc.edu (C.S.C.); 2UAB Center for Exercise Medicine, University of Alabama at Birmingham, Birmingham, AL 35294, USA; 3Nathan Shock Center, University of Alabama at Birmingham, Birmingham, AL 35294, USA; 4CardioRenal Physiology and Medicine Section, Division of Nephrology, Department of Medicine, University of Alabama at Birmingham, Birmingham, AL 35294, USA; kerikemp@uab.edu; 5Center for Cognitive Aging and Memory, Department of Neuroscience and McKnight Brain Institute, College of Medicine, University of Florida, Gainesville, FL 32611, USA; burkes@ufl.edu; 6Integrative Center for Aging Research, University of Alabama at Birmingham, Birmingham, AL 35294, USA; 7Geriatric Research Education and Clinical Center, Birmingham VA Medical Center, Birmingham, AL 35294, USA

**Keywords:** ketogenic diet, intermittent fasting, gut, 16S, cytokine

## Abstract

Both ketogenic diets (KD) and time-restricted feeding (TRF) regimens have the ability to influence several parameters of physical health, including gut microbiome composition and circulating cytokine concentration. Moreover, both of these dietary interventions prevent common impairments associated with the aging process. However, significantly altering macronutrient intake, which is required for a KD, may be unappealing to individuals and decrease compliance to dietary treatments. In contrast to a KD, TRF allows individuals to continue eating the foods they are used to, and only requires a change in the time of day at which they eat. Therefore, we investigated both a KD and a diet with a more Western-like macronutrient profile in the context of TRF, and compared both diets to animals allowed access to standard chow ad libitum in young adult and aged rats. While limited effects on cytokine levels were observed, both methods of microbiome analysis (16S sequencing and metagenomics) indicate that TRF and KDs significantly altered the gut microbiome in aged rats. These changes were largely dependent on changes to feeding paradigm (TRF vs. ad libitum) alone regardless of macronutrient content for many gut microbiota, but there were also macronutrient-specific changes. Specifically, functional analysis indicates significant differences in several pathways, including those involved in the tricarboxylic acid (TCA) cycle, carbohydrate metabolism and neurodegenerative disease. These data indicate that age- and disease-related gut dysbiosis may be ameliorated through the use of TRF with both standard diets and KDs.

## 1. Introduction

While extensive work indicates that ketogenic diets (KD) improve several markers of health, long term compliance can be difficult to achieve [1,2,3]. This is an important barrier to a longer health span, as adherence to a higher-quality diet increases the likelihood of successful aging [4]. Time-restricted feeding (TRF; also referred to as intermittent fasting) confines the period during which food is consumed to specific hours of the day (typically a 6–8 h window). TRF or intermittent fasting both refer to a variety of dietary paradigms, each of which incorporates short periods of fasting ranging from hours to a whole day, with many options therein [5]. However, unlike a KD, the macronutrient composition of the diet is not altered. Compliance with TRF is generally higher, even in older adults [6], who typically struggle with various dietary interventions that are more restrictive in nature, such as those that restrict carbohydrates or restrict feeding to a certain number of calories per day [7,8]. Therefore, using a well-characterized rodent model of aging, the Fisher344 x Brown Norway (FBN) rat, we investigated both a KD and a diet with a more Western-like macronutrient profile in the context of time-restricted feeding (TRF), and compared both diets with animals allowed access to standard chow ad libitum. The KD was given as a TRF diet to delineate differences in changing macronutrient composition from changes in consumption patterns, as it is already well established that KDs themselves influence peripheral health and gut microbiome composition [9,10,11,12,13,14].

TRF may provide an alternative dietary paradigm providing similar benefits to KDs regarding gut microbiome changes, without requiring large shifts in macronutrient consumption, making them more easily translatable and maintainable for humans. Advanced age is associated with changes in the composition and density of the gut microbiome [15], known as dysbiosis, which can negatively influence peripheral health and quality of life [16]. Recent evidence has shown that TRF not only results in significant weight loss in obese individuals, but is also able to improve several measures of physiological function, including insulin regulation [17]. Additional improvements include improved mitochondrial function in older individuals [18], as well as delayed tumor onset and reduction in tumor-like lesion quantity [19]. Moreover, TRF decreases the circulating levels of pro-inflammatory cytokines, including tumor necrosis factor alpha (TNFα) and interleukin 1 beta (IL-1β) [20].

While modulating dietary macronutrients, such as a KD, is known to have a profound impact on microbiome composition [9,10,11,12,13,14], significantly less is known about the impact of TRF on microbial composition and diversity. While one group reports no significant differences in alpha diversity or phylum-level abundance in obese humans undergoing TRF for 12 weeks despite improved body condition, this group did not offer additional analyses at lower taxonomic levels nor were beta diversity measures explored, leaving many questions on the impact of TRF on gut microbiome composition unanswered [21]. A second group investigating TRF in healthy, young adult males did find that gut microbial richness was significantly enhanced and microbial community makeup differed post dietary intervention [22,23]. Yet a third group demonstrated that the highly dynamic nature of the gut microbiome includes daily cyclical fluctuations in composition, which can be altered by TRF [24], and may account for differences across studies. The influence of TRF on the gut milieu of aged individuals, however, remains under investigated.

Several animal models of disease have demonstrated the utility of utilizing TRF to manipulate the gut microbiome and ameliorate or prevent physiological decline, though these studies are limited in number and do not encompass the full treatment potential of gut microbiome manipulations. For example, TRF resulted in significantly longer survival and reduction in diabetic retinopathy end points through altered gut microbiome composition (increased Firmicutes, Bacteroidetes and Verrucomicrobia) in the db/db diabetes mouse model [25]. Moreover, TRF in these mice improved microbial metabolites related to cognitive function, an affect ameliorated by antibiotic administration [26]. These data demonstrate the strong link between gut microbiome structure and distal organ systems, including the central nervous system. Moreover, in the autoimmune encephalomyelitis (EAE) mouse model of multiple sclerosis (MS), TRF (or even receipt of the microbiome taken from a TRF mouse) increased microbial diversity, resulting in altered metabolic pathway function and correlating with the amelioration of MS pathology [27]. TRF alterations in gut microbiome composition also mediated the reduction in blood pressure in spontaneously hypertensive stroke-prone rats [28]. Additionally, the large role gut health and microbiome composition play in energy homeostasis and metabolic function greatly influences cognitive performance [29,30,31]. Therefore, the gut may be one potential avenue through which we can target neurobiological health, along the gut–brain axis [32,33] through novel therapeutic interventions. While there is extremely limited information on the combination of TRF with a KD in humans, limited data from case studies [34] and small clinical trials [35] indicate that the combination may provide therapeutic relief for uncontrollable diabetes or insulin resistance. However, there were notable differences in seizure relief in children who differed in response to the two dietary paradigms [36].

Interest in the effects of TRF on gut microbiome composition and health extend beyond those interested in weight loss or therapeutic intervention. For example, individuals who observe Ramadan, during which the participants fast from sun up to sun down for 29 days, exhibit increased alpha and beta diversity, as well as changes in the abundance of several major phyla [37,38,39]. Specifically, alterations in *Lachnospiraceae* abundance were observed in one of these groups, which may help prevent some of the deleterious aspects of the aging process through enhancing butyrate production [40].

The gut microbiome greatly influences several systems, including neurological function, cardiovascular function, immune system function and more. Therefore, to better investigate the differential, or possibly synergistic, effects of KDs and TRF on physiological parameters, we collected fecal samples from animals that underwent each dietary paradigm (TRF Keto, TRF Control, and ad libitum rodent chow) and assessed changes in microbial composition through 16S sequencing. Chronic delivery of these diets (28 weeks) was used to prevent alterations due to dietary acclimation. Additional metagenomic analyses were utilized to assess potential functional outcomes of observed microbial shifts.

## 2. Materials and Methods

### 2.1. Subjects

A total of 20 young (5 months) and 13 aged (21 months) male Fisher 344 x Brown Norway hybrid F1 rats were used in this study. Rats were split across 3 diet groups: one group (ad lib; 6 young and 4 aged) was given ad libitum access to standard rodent chow (18% fat, 24% protein, 58% carbohydrate; Envigo, Teklad 2918), a second group (TRF Keto; 7 young and 4 aged) was given a ketogenic diet (75.85% fat, 20.12% protein, 3.85% carbohydrate; Lab Supply; 5722, Fort Worth, TX, USA; see [3,41] for additional diet details) fed to them once daily (time-restricted feeding; TRF) and the third group (TRF Control; 7 young and 5 aged) was given a diet similar in macronutrient ratio to the control diet, but also fed to them once daily (16.35% fat, 18.76% protein, 64.89% carbohydrate; Lab Supply; 1810727, Fort Worth, TX, USA; see [3,41] for additional diet details). Rats in both TRF groups were fed approximately 7 h after the onset of their dark (active) phase, which may be considered comparable to late-TRF [42]. Each group remained on their respective diets for 7 months (28 weeks).

### 2.2. Tissue Collection

At the time of euthanasia, rats were placed in a bell jar containing isoflurane-saturated cotton (Abbott Laboratories, Chicago, IL, USA) until righting reflex was lost. Rats were immediately decapitated and tissue was immediately extracted. Trunk blood was collected, allowed to sit at room temperature for at least 20 min and then spun at 4 °C. Supernatant was collected and stored at −80 °C until use. Fecal samples were collected directly from the distal colon, placed in Para-Pak (Meridian Bioscience Inc., Cincinnati, OH, USA) and immediately stored at −80 °C until use.

### 2.3. Nutritional Ketosis

Nutritional ketosis was determined via circulating glucose (mg/dL) and ketone body (β-hydroxybutyrate; BHB; mmol/L)) concentration using the Precision Xtra blood monitoring system (Abbott Diabetes Care, Inc, Alameda, CA, USA). These values were utilized to calculate a glucose ketone index (GKI), in which lower values indicate a higher level of nutritional ketosis, using the formula: glucose (mgdL)18.016BHB (mmolL).

### 2.4. Cytokine Analysis

Circulating concentration of the cytokines TNFα, interferon gamma (IFN-γ) and interleukins (ILs) 4, 6 and 10 were quantified from blood samples utilizing a Meso Scale Discovery (MSD; Rockville, MD, USA) Rat Proinflammatory Panel 2 and a Quick Plex SQ 120 imager (MSD, Rockville, MD) using electrochemiluminescence technology.

### 2.5. Fecal Microbiome Taxonomy

Analysis of fecal microbiome was performed via 16S rRNA gene sequencing as previously described [43,44]. Briefly, a Zymo Research Fecal DNA isolation kit (Zymo Research, Irvine, CA, USA; catalog # D6010) was used for DNA isolation. Isolated DNA was then quantitated and barcoded via polymerase chain reaction (PCR) amplification of the V4 region of the 16S rRNA gene [45] using degenerate primers with slight modifications as described [46] from the original Capaoraso primer sequences [47]. As previously described [46,48], PCR products were resolved on agarose gels; DNA was isolated and purified using Qiagen kits (Qiagen, Hilden, Germany); and then quantitated. The products were sequenced on the Illumina MiSeq platform (Illumina, San Diego, CA, USA), a single-flow cell, single-lane instrument.

Following quality control, exact amplicon sequence variants (ASVs) were resolved and taxonomy was assigned using the SILVA small subunit ribosomal RNA database version 132 (Max Planck Institute for Marine Microbiology and Jacobs University, Bremen, Germany) [49]. Alpha diversity, or the diversity within samples, was quantified for age, diet group and feeding paradigm utilizing the microbiome package in R [50] with the following measures: Richness was evaluated utilizing the Chao1 index [51], which estimates the number or species in a community. Diversity was evaluated using the inverse Simpson index, which measures the dominances of a multispecies community [52]. Evenness was then determined utilizing Simpson’s index, which accounts for the number of species present as well as relative abundance of each of those species. Dominance was also found using Shannon’s index, as the dominance index gives the abundance of the most abundant species transformation [53]. The rarity index characterizes the concentration of species at low abundance, using the skewness of the frequency distribution of arithmetic abundance classes [54] and using the log-modulo transformation [53]. Beta diversity was determined via permutational multivariate analysis of variance (PERMANOVA) on Bray–Curtis, weighted Unifrac [55], and unweighted Unifrac distance matrices using the Phyloseq package [56] in R [57]. Analysis of Compositions of Microbiomes (ANCOM) with Bias Correction was used to test for differential abundance at several levels across diet and feeding paradigm treatment groups using modified versions of previously published ANCOM scripts [58,59]. Briefly, raw counts were filtered for any sequence present in at least 30% of all samples. The ANCOM detection limit was set to the default value of 0.7 and was run on centered log ratio transformed (CLR) count data using a Benjamini–Hochberg-corrected significance level of 0.05 and adjusting for cohort grouping as a covariant.

### 2.6. Metagenomics

Metagenome libraries were generated by using the Illumina NextSeq 500 sequencer platform (Illumina, San Diego, CA, USA) on 4 aged samples per diet group. Quality control was performed using the MG-RAST v4.0.3 pipeline quality control filter [60], which resulted in an average of 24,425,398 reads to be included in further analysis. Of those, ~55% of the proteins were annotated to an assigned function or specific gene by the MG-RAST v4.0.3 pipeline.

### 2.7. Statistical Analysis

Cytokine, GKI, and alpha diversity data were analyzed with two-way ANOVAs, with the between-subjects factors of age and diet. Post hoc analyses were conducted via Tukey’s multiple comparisons test, when appropriate. T-tests were utilized for post hoc analyses when statistically indicated via main effect of any between-subjects variable. See microbiome and metagenomics methods above for related statistical analysis. All analyses were performed with GraphPad Prism v9.2.0 (GraphPad Software, San Diego, CA USA), R v4.1.1, or IBM SPSS v25 (SPSS Inc., Chicago, IL, USA). Statistical significance was considered at p values less than 0.05, unless the Benjamini–Hochberg method [61] of correcting for false discovery rate was applied as stated within the text.

## 3. Results

### 3.1. Confirmation of Nutritional Ketosis

At the time of euthanasia, circulating glucose and ketone body levels were quantified to calculate a glucose ketone index (GKI), in which lower values indicate a higher level of nutritional ketosis (Figure 1). While there were no effects of age on GKI (F_(1,27)_ = 0.66; *p* = 0.42), there was a significant effect of diet (F_(2,27)_ = 44.24; *p* < 0.0001), as both the ad libitum (T_(27)_ = 7.12; *p* < 0.0001), and TRF Control-fed (T_(27)_ = 8.92; *p* < 0.0001) rats had significantly higher GKI levels than TRF Keto-fed rats. Ad libitum and TRF Control-fed rats did not have significantly different GKI levels (T_(27)_ = 1.43; *p* = 0.42).

### 3.2. Cytokine Analysis

Circulating cytokine concentrations were quantified in serum collected at the time of sacrifice. For all five cytokines assessed, there were no significant effects of age, though there was a strong trend for an increase with age in TNFα (F_(1,27)_ = 4.05, *p* = 0.05; *p* > 0.24 for all others). Moreover, there were no effects of diet on any of the cytokines aside from TNFα (F_(2,27)_ = 3.63, *p* = 0.04; *p* > 0.40 for all others). Post hoc analysis for TNFα revealed that while aged rats fed ad libitum had significantly higher cytokine levels than young rats (t_(29)_ = 2.52, *p* = 0.03), this was not the case for TRF-fed rats (t_(29)_ = 0.69, *p* > 0.99) regardless of macronutrient profile. There were no significant interactions between age and diet for any cytokines (*p* > 0.26 for all comparisons).

### 3.3. 16S Microbiome Analysis

#### 3.3.1. Diversity

Alpha diversity was calculated through several common measures, as described above. The results of two-way ANOVAs with the between-subjects factors of age and diet group, with post hoc assessment of age within each diet group are displayed in Table 1. Moreover, this table also includes the results of two-way ANOVAs with the between-subjects factors of age and feeding paradigm (with both TRF groups combined), with post hoc assessment of feeding paradigm within each age group. The Inverse Simpson measure of diversity was the only measure indicating significant differences across feeding paradigm (F_(1,29)_ = 26.28; *p* < 0.0001; Figure 2).

Beta diversity was calculated using the Bray–Curtis (BC) Dissimilarity, unweighted Unifrac (UWU) and weighted Unifrac (WU) analyses (Figure 3). For each of these methods, a PERMANOVA comparing all three diet groups indicated a significant effect of diet (BC: F_(2,30)_ = 5.94; *p* = 0.001; UWU: F_(2,30)_
^=^ 4.96; *p* = 0.001; WU: F_(2,30)_ = 8.24; *p* = 0.001). Similarly, a PERMANOVA comparing the ad libitum to TRF feeding paradigms also revealed a significant effect of feeding paradigm on beta diversity (BC: F_(1,31)_ = 5.23; *p* = 0.001; UWUF_(1,31)_ = 5.53; *p* = 0.001; WU: F_(1,31)_ = 5.55; *p* = 0.001).

#### 3.3.2. Differential Abundance

Analysis of composition of microbiomes (ANCOM) was utilized to examine taxa that had statistically different abundance between dietary (Figure 4A) and feeding paradigm groups (Figure 4B), each adjusted for age. At the phylum level, five phyla (Actinobacteria, Cyanobacteria, Deferribacteres, Patscibacteria and Verrucomicrobia) are significantly influenced by diet (Figure 4C) and three phyla (Actinobacteria, Patscibacteria and Verrucomicrobia) are significantly influenced by feeding paradigm (Figure 4D).

ANCOM was also repeated at the genus level, which demonstrated that 22 genera were significantly altered by diet group, and 9 genera were significantly altered by feeding paradigm (Figure 5). Interestingly, all genera significantly affected by feeding paradigm were also significantly altered by diet group. The difference in centered log ratio (CLR)-transformed counts utilized in the ANCOM analysis between each TRF group and the ad libitum group were plotted against the W statistic, which represents the number of times the ratio between the given taxon and each of the other taxa in the community was significantly different between the two treatment groups (Figure 5B,C).

#### 3.3.3. Metagenomics

In addition to taxonomical differences, metagenomic analyses were used to identify functional differences (Figure 6). Community-wide functional representation was assessed by mapping metagenomic reads to the Kyoto Encyclopedia of Genes and Genomes (KEGG) Orthology (KO) database via MG-RAST. Within each level 2 subgroup, level 3 groupings were investigated via two-way ANOVA with diet group and level 3 classification as between-subjects factors. Based on our a priori hypothesis that metagenomic data would significantly differ based on feeding paradigm regardless of macronutrient composition, post hoc analysis was performed for each TRF diet group relative to ad libitum rats using Dunnett’s Multiple Comparisons. Significant differences at level 3 were then further investigated at the functional level in pertinent pathways.

Several KEGG pathways involved in metabolic processes were altered by both dietary interventions. Firstly, within the carbohydrate metabolism level, components of the tricarboxylic acid (TCA) cycle were significantly upregulated by TRF relative to ad libitum, including citrate synthase (CS), pyruvate carboxylase subunit B (pycB), 2-oxoglutarate ferredoxin oxidoreductase subunits alpha and beta (korA and korB), and malate dehydrogenase (mdh). Secondly, amino acid metabolic pathways were also altered by TRF, including glycine, serine and threonine metabolism and alanine, aspartate and glutamate metabolism. Within alanine, aspartate and glutamate metabolism, several pathways were specifically upregulated by TRF Keto, but not TRF Control, including asparagine synthase (asnB), carbamoyl-phosphate synthase large subunit (carB), glutamate dehydrogenase (NADP+; gdhA), amidophosphoribosyltransferase (purF) and glutamate synthase (NADPH/NADH) large and small chains (gltB and gltD). Only one pathway was significantly upregulated by TRF Control but not TRF Keto, alanine dehydrogenase (ald). Within the glycine, serine and threonine metabolism level, dihydrolipoamide dehydrogenase (DLD), glycine dehydrogenase (GLDC) and glycine C-acetyltransferase (GCAT) were all upregulated by both TRF diets. However, ilvA, tdcB; threonine dehydratase (ilvA), 2,3-bisphosphoglycerate-dependent phosphoglycerate mutase (PGAM), tryptophan synthase beta chain (trpB) and phosphoserine aminotransferase (serC) were significantly upregulated by TRF Keto alone. Moreover, aminomethyltransferase (gcvT) and bifunctional aspartokinase/homoserine dehydrogenase 1 (thrA) associated pathways were significantly upregulated by TRF Control alone and glycine hydroxymethyltransferase (glyA) was significantly downregulated by TRF Control alone. Thirdly, within the energy metabolism level, oxidative phosphorylation was significantly upregulated by the TRF Keto, but not TRF Control, demonstrating that not all effects are due to feeding paradigm alone.

Relatedly, within the environmental information processing level, signal transduction along the PI3K–Akt pathway was significantly upregulated by TRF Keto, but not TRF Control. This pathway is vital to cellular and organismal function, as it is involved in cell proliferation and survival by phosphorylating (and thereby modulating) a variety of substrates. This includes, but is not limited to, glycogen synthase kinase-3 (GSK3), GLUT4 (resulting in translocation to the membrane), cyclin-dependent kinase inhibitors, P21/Waf1/Cip1 and P27/Kip2 and mammalian target of rapamycin (mTOR) [62]. Conversely, several neuroactive ligand–receptor interactions were significantly downregulated in both TRF-fed diets relative to ad libitum. Both TRF diets significantly downregulated several receptors and receptor subunits, including formyl peptide receptor-like (FPRL), gamma-aminobutyric acid receptor subunit epsilon (GABRE), metabotropic glutamate receptor 6/7/8 (GRM_6_7_8), 5-hydroxytryptamine receptor 1 (HTR1), trypsin (PRSS), neuropeptide Y receptor type 1/4/6 (NPY1R_4R_6R) and trace amine associated receptor (TAAR). Two additional receptors, the benzodiazepine receptor (BZRP) and glutamate [NMDA] receptor subunit epsilon-2 (GRIN2B), were significantly lowered by TRF Control only; and one, prostaglandin D2 receptor (PTGDR), by TRF Keto only.

Within the human diseases level, pathways associated with neurodegenerative diseases, in particular Alzheimer’s disease, were significantly downregulated following TRF. Interestingly, the pathway associated with amyloid beta A4 protein (APP) was downregulated by TRF Keto only, as was F-type H^+^-transporting ATPase subunit alpha (ATPeF1a) and NADH dehydrogenase (ubiquinone) 1 beta subcomplex 4 (NDUFB4), though NADH dehydrogenase (ubiquinone) 1 beta subcomplex 3 (NDUFB3) was significantly affected by both TRF groups. Both TRF groups also significantly reduced pathways associated with F-type H+-transporting ATPase subunit a (ATPeF0A), insulin-degrading enzyme (IDE), and cytochrome c oxidase subunit 1 (COX1), though cytochrome c oxidase subunit 3 (COX3) was influenced by TRF Control only.

At the systemic level, protein digestion and absorption were significantly influenced by the diets. Specifically, the pathway associated with dipeptidyl-peptidase 4 (DPP4) was upregulated by both TRF diets. Two main areas of transport and catabolism within the cellular processes level were significantly upregulated by both TRF diets: the peroxisome and lysosomes. Long-chain acyl-CoA (ACSL) and catalase (CAT), both associated with the peroxisome, and glucosylceramidase (GBA), hexosaminidase (HEXA_B) and sialidase-1 (NEU1), associated with the lysosome, were all significantly upregulated by both TRF diets.

## 4. Discussion

In this cohort of male FBN rats, it appears that feeding paradigm (TRF vs. ad libitum) had a significant effect on gut microbiome composition, in addition to macronutrient composition. While others have observed differences in gut microbiome composition following ketogenic diets (KD; reviewed in [63]), we aimed to both replicate this finding as well as expand upon the dearth of information regarding TRF and gut microbiota composition (GMC).

Our data indicate that while the level of most cytokines remained stable throughout adulthood, aged rats randomized to consume chow ad libitum demonstrated excessive TNFα levels in circulation. This effect was prevented in the rats consuming either the KD or CD with TRF. However, significant differences were observed in microbial diversity across both diet groups (i.e., differing macronutrient ratio) and feeding paradigm groups (i.e., TRF vs. ad libitum consumption).

Both diet group and feeding paradigm yielded significantly different GMC. Moreover, differential abundance analysis revealed 22 genera were significantly altered by diet group, and 9 genera were significantly altered by feeding paradigm. Gut microbiome (GMB) diversity assessed by Simpson’s inverse index, which quantifies biodiversity by taking into account richness and evenness, was increased by TRF compared to ad libitum feeding. Consumption of either the TRF Keto or TRF Control diets prevented elevated TNFα levels in circulation that occurred in aged rats consuming the ad libitum diet. This indicates that these changes in GMB diversity and composition documented with TRF feeding paradigm may alter the proinflammatory status in older animals. While it is not surprising that diet macronutrient composition altered the composition and function of the GCM, it is interesting that a TRF feeding paradigm had strong effects on GCM diversity and composition.

Our microbiome analysis supports the efficacy of TRF to significantly alter metabolite production and utilization. Specifically, altering food consumption pattern in this study resulted in changes in gut microbes associated with short-chain fatty acid (SCFA) production. These changes are interesting in light of recent work demonstrating that over-production of gut microbiota-produced acetate (GMPA) leads to insulin over-secretion and obesity symptoms [64]. This accumulation of acetate, which occurs following consumption of a high-fat diet, results in parasympathetic nervous system activation and over-secretion of insulin and ghrelin. Conversely, the accumulation of gut microbiota-produced butyrate (GMPB), but not GMPA, enhances AMPK signaling, reducing expression of lipogenesis-associated genes and triggering insulin sensitivity [65]. Thus, a shift from acetate-producing to butyrate-producing bacteria may improve obesity-related gut dysbiosis and metabolic health.

*Akkermansia muciniphila* has recently been described as “a next-generation beneficial microbe” [66]. Our data indicate that TRF enriches for the genus *Akkermansia*, though increases in abundance only reached significance for rats also consuming a TRF Keto, and not the TRF Control, relative to ad libitum-fed rats. *Akkermansia* influences many aspects of metabolic health, including glucose metabolism, lipid metabolism, and intestinal immunity. The anti-diabetes drug Metformin enhances *Akkermansia* levels [67], which is one possible mechanism by which Metformin appears to be protective against cognitive decline in aged subjects with type II diabetes [68]. Moreover, *Akkermansia muciniphila* is necessary for the anti-seizure effects of the KD [11], strongly demonstrating the link between neurobiological function and gut microbiome composition.

In addition to *Akkermansia muciniphila*, there are many ways by which the gut microbiota influence cognitive function. Our metagenomic analysis revealed that in addition to metabolic and other cellular processes, several neurobiological pathways were affected by the diets utilized herein. Firstly, pathways involved in the development of neurodegenerative diseases were significantly different across groups. The pathway associated with amyloid beta A4 protein (APP) production was significantly reduced in TRF Keto-fed rats relative to ad libitum-fed rats, but not TRF Control-fed rats. While the full range of functions of APP remains unknown, cleavage of APP generates neurotoxic β-amyloid peptide (Aβ), which accumulates within the brains of individuals with Alzheimer’s disease (AD) [69]. KDs have been suggested to be efficacious in the treatment of AD by numerous groups [70,71,72,73] and recently reviewed in [68] and our data indicate that a potential mechanism by which it is influencing brain pathology may be altered gut microbiome composition. Additionally, mitochondrial dysfunction, particularly defects in cytochrome C oxidase (COX), is widely reported within AD [74,75]. Our metagenomic analysis indicates significant alterations in COX subunits 1 and 3 with both TRF diets and TRF Control only, respectively. Moreover, TRF Keto significantly altered ATPase activity through subunit a, which would also affect a cell’s ability to function normally. Insulin-degrading enzyme (IDE) strongly links metabolic function with AD, as it not only degrades insulin but also Aβ [76], and was significantly decreased following TRF in our dataset.

Secondly, many pathways related to neuroactive ligand signaling were significantly decreased following TRF. This includes pathways related to gamma-aminobutyric acid receptor subunit pi (GABRP), melanocortin 3 receptor (MC3R), nicotinic acetylcholine receptor gamma (CHRNG), glycine receptor beta (GLRB) and prostaglandin E receptor 2 (PTGER2). However, whether these changes are limited to localized effects or influence neurotransmitter signaling processes within the brain remains to be determined.

Thirdly, several key pathways involved in the tricarboxylic acid (TCA) cycle were significantly upregulated following the TRF diets. Specifically, malate dehydrogenase (mdh) and 2-oxoglutarate ferredoxin oxidoreductase subunits alpha and beta (korA and korB) were significantly upregulated by both TRF diets, fumarate hydratase class I (fumA and fumB) and citrate synthase (CS) were upregulated by TRF Keto only and pyruvate carboxylase subunit B (pycB) was upregulated by TRF Control only. Relatedly, other pathways involved in carbohydrate metabolism were significantly altered following dietary intervention, which can influence TCA cycle function. Several aspects of both the alanine, aspartate and glutamate metabolic pathway and the glycine, serine and threonine metabolic pathway were significantly altered by one or more of the TRF diets. Some of these aforementioned changes in the GMB following TRF may begin to provide mechanistic explanation for the plethora of mental health changes (including depression, cognitive impairment, sleep disorders, dementia and AD) following TRF, as reviewed in [77].

One caveat to this study, which likely prevented any age-related differences in microbial composition, is that the ‘young’ rats were fully developed adults at the onset of this study, and the chronic nature of the dietary paradigm resulted in rats being closer to middle aged (13 months of age) at the time of sample collection. Future studies initiating diet interventions at earlier time points are needed to resolve age-related differences in the efficacy of TRF paradigms versus macronutrient composition. Moreover, chronic dietary interventions such as this one may have lower translational potential and the utility of other paradigms, such as cycling on and off the ketogenic diet, may be better targets for future studies [78,79].

There are several possible mechanisms by which TRF would change GMB composition. Firstly, TRF elicits some amount of ketone body production in the liver as soon as 8 h after beginning a fast [80], providing different nutrient sources than typical eating paradigms. Altered nutrient sources for gut microbes significantly influence production of many metabolites, including neurotransmitters and other key enzymes [81,82]. The degree to which ketones directly influence health outcomes via the gut microbiota remains largely unknown, but likely correlates with the abundance of ketotic- vs. glucose-related fuel sources in the gut. In fact, recent work demonstrated that ketone production by KD and TRF affected the gut microbiota and disease progression differently in a rat model of AD [83]. Secondly, the timing of food consumption modulates circadian rhythms, which then modulates a host of other physiological functions [84]. This includes the gut microbiota, which undergoes significant restructuring throughout the day [85]. Moreover, this leads to fluctuations in the concentration of key bacterial metabolites, including acetate, propionate and butyrate [86]. Thirdly, TRF is often, though not always, accompanied by weight loss. Obese individuals have significantly altered gut microbiota [87], thus restoration of a more lean phenotype may restore obesity-related gut dysbiosis. Moreover, TRF promotes browning of adipose tissue, which may further decrease obesity and/or alter GCM composition [88], though a combination of TRF and Keto reduces brown adipose tissue volume [41].

## 5. Conclusions

Both macronutrient composition and feeding paradigm result in significant alteration in gut microbiome composition. These changes may mediate improved physiological and cognitive outcomes following dietary implementation, both in healthy individuals and in disease. Moreover, our data indicate that this could be equally effective in older populations, who experience greater cognitive decline and higher incidence of metabolic and neurological disorders. Given the lack of evidence using pharmaceutical interventions to prevent aging and age-related disease, this new concept of manipulating the gut peripherally to target distal organ health and function may serve as both a first line of evidence-based, low-risk intervention for health care providers seeking to increase patients’ health span, as well as an adjuvant to clinical therapies, through relatively simple dietary intervention. Our data indicate that both types of dietary interventions are capable of altering the gut microbiome in different ways, and selection of dietary interventions may rely on both physiological outcomes and patient preference.

## Figures and Tables

**Figure 1 nutrients-14-01758-f001:**
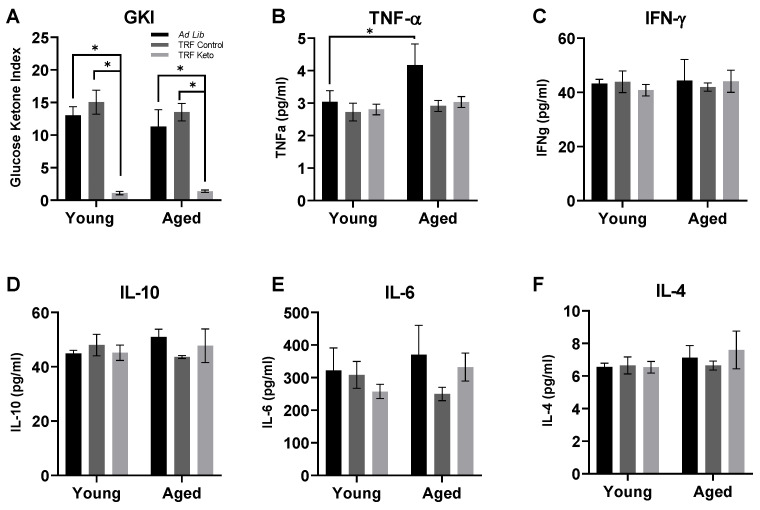
Circulating biomarkers following dietary intervention. (**A**) The glucose ketone index (GKI) was significantly lower in Time restricted fed (TRF) Keto-fed rats than in either ad libitum or TRF Control-fed rats. (**B**–**F**) Levels of circulating cytokines remained largely unaffected by either age or diet, with the exception of TNFα, for which a significant effect of diet (*p* = 0.04) was found. Abbreviations: TNFα: tumor necrosis factor alpha; IFN-γ: interferon gamma; IL4/6/10: interleukin 4/6/10. All values represent the mean ± the standard error of the mean (SEM), * indicates *p* < 0.05.

**Figure 2 nutrients-14-01758-f002:**
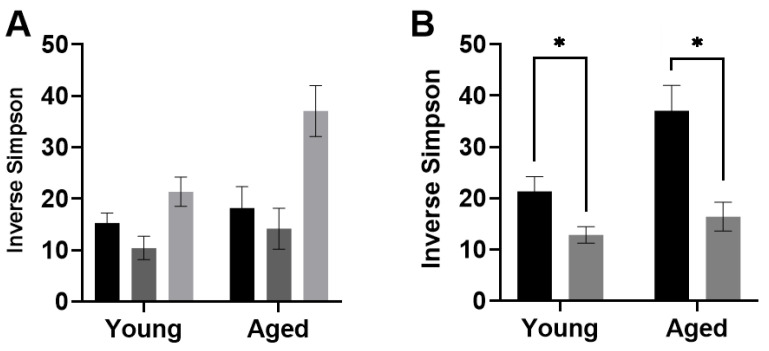
Alpha diversity by (**A**) diet group and (**B**) feeding paradigm (TRF vs. ad libitum regardless of macronutrient ratio) suggests that feeding paradigm more strongly influences microbial diversity than altered macronutrient ratio alone. All values represent the mean ± the standard error of the mean (SEM), * indicates *p* < 0.05.

**Figure 3 nutrients-14-01758-f003:**
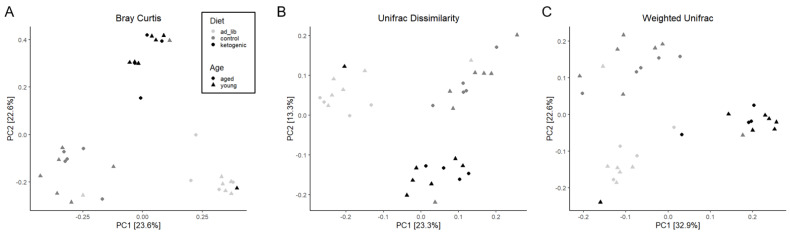
Beta diversity across diet and age groups. All three methods utilized, which includes the (**A**) Bray–Curtis Dissimilarity, (**B**) unweighted Unifrac Dissimilarity and (**C**) weighted Unifrac Dissimilarity demonstrated significantly different beta diversities based on diet and feeding paradigm (TRF versus ad libitum) groups. PC: Principle Component.

**Figure 4 nutrients-14-01758-f004:**
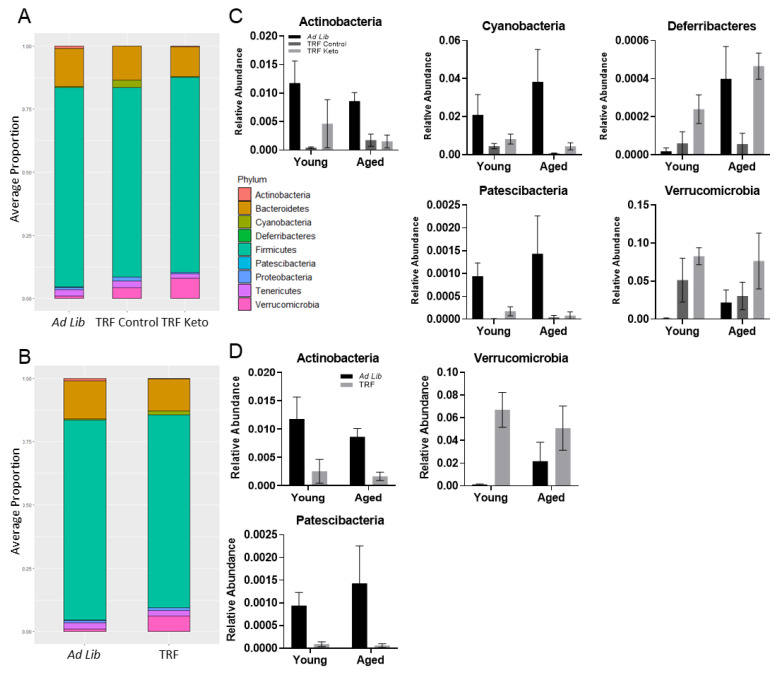
Diet and feeding paradigm influence on gut microbe abundance at the phylum taxonomic level. Relative abundance at the phylum taxonomic level (**A**) by diet group and (**B**) feeding paradigm (TRF versus ad libitum). Significantly different phyla are shown by (**C**) diet group and (**D**) feeding paradigm. All values in C–D represent the mean ± the standard error of the mean (SEM).

**Figure 5 nutrients-14-01758-f005:**
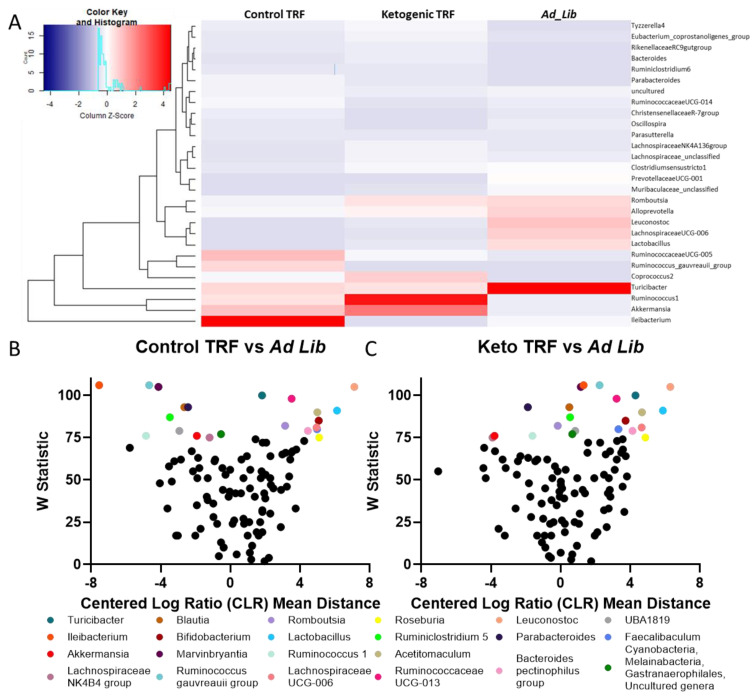
Diet and feeding paradigm (TRF versus ad libitum) influences on gut microbe abundance at the genus taxonomic level. (**A**) Heat map of genera relative abundance by diet group. Analysis of composition of microbiomes (ANCOM) differential abundance volcano plots at the bacterial genus level for (**B**) TRF Control and (**C**) TRF Keto relative to ad libitum-fed rats. ANCOM analysis utilized the centered log ratio (CLR)-transformed ASV count table. Only significantly different genera are colored, non-significant taxa are displayed in black.

**Figure 6 nutrients-14-01758-f006:**
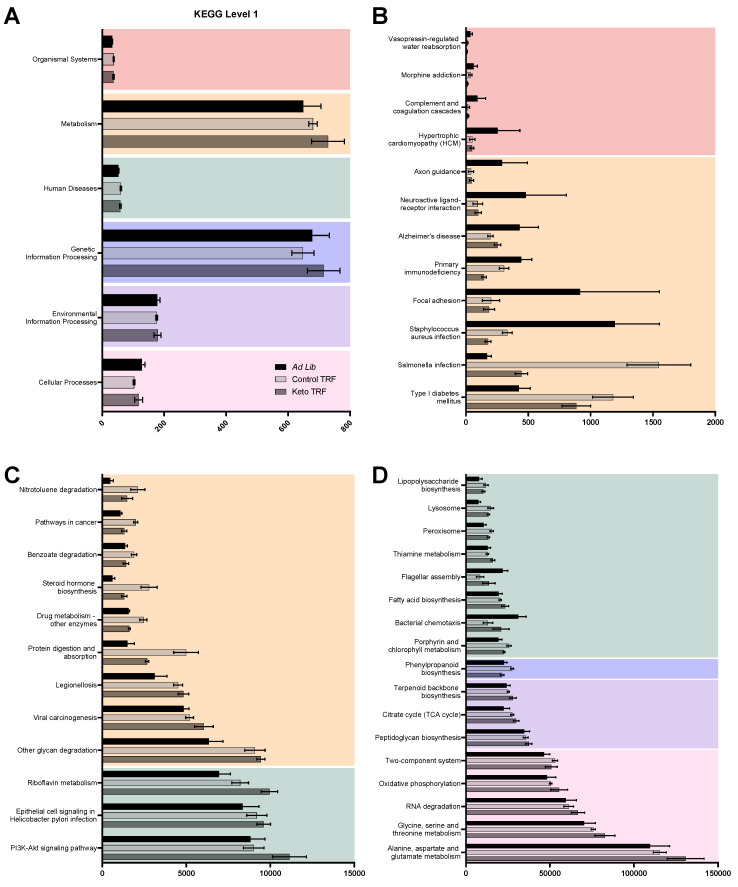
Cluster of Kyoto Encyclopedia of Genes and Genomes (KEGG)-based annotation analysis by diet group. (**A**) Level 1 KEGG classification relative abundance across diet groups. (**B**–**D**) Within each level 2 subgroup, level 3 groupings were investigated and significantly affected level 3 classifications are displayed here.

**Table 1 nutrients-14-01758-t001:** Two-way ANOVAs with the between-subjects factors of age and diet group, with post hoc assessment of age within each diet group on five distinct alpha diversity measures.

	3 Diet Comparison x Age							Feeding Paradigm x Age						
		F (df, Error)	*p*		t	DF	*p*		F (df, Error)	*p*		t	DF	*p*
**Chao1**	**Interaction**	F (2, 27) = 1.339	0.2791	**TRF Control**	0.461	27	0.9566	**Interaction**	F (1, 29) = 2.432	0.1297	**Young**	1.448	29	0.2917
	**Age**	F (1, 27) = 1.211	0.2808	**TRF Keto**	0.4362	27	0.9628	**Age**	F (1, 29) = 2.512	0.1238	**Aged**	3.185	29	0.0069
	**Diet**	F (2, 27) = 5.507	0.0099	**Ad libitum Chow**	1.836	27	0.2147	**Feeding Paradigm**	F (1, 29) = 11.46	0.0021				
**Diversity (Inverse Simpson)**	**Interaction**	F (2, 27) = 2.385	0.1112	**TRF Control**	1.413	27	0.426	**Interaction**	F (1, 29) = 4.531	0.0419	**Young**	2.378	29	0.0479
	**Age**	F (1, 27) = 8.091	0.0084	**TRF Keto**	3.909	27	0.0017	**Age**	F (1, 29) = 11.44	0.0021	**Aged**	4.673	29	0.0001
	**Diet**	F (2, 27) = 14.15	<0.0001	**Ad libitum Chow**	5.134	27	<0.0001	**Feeding Paradigm**	F (1, 29) = 26.28	<0.0001				
**Evenness (Simpson)**	**Interaction**	F (2, 27) = 0.2374	0.7903	**TRF Control**	1.469	27	0.393	**Interaction**	F (1, 29) = 0.3164	0.5781	**Young**	1.374	29	0.3276
	**Age**	F (1, 27) = 1.989	0.1699	**TRF Keto**	1.367	27	0.4544	**Age**	F (1, 29) = 2.327	0.138	**Aged**	1.84	29	0.1462
	**Diet**	F (2, 27) = 3.718	0.0375	**Ad libitum Chow**	2.725	27	0.0331	**Feeding Paradigm**	F (1, 29) = 5.265	0.0292				
**Dominance (Simpson)**	**Interaction**	F (2, 27) = 0.4912	0.6173	**TRF Control**	2.065	27	0.1388	**Interaction**	F (1, 29) = 0.009902	0.9214	**Young**	2.089	29	0.089
	**Age**	F (1, 27) = 1.947	0.1743	**TRF Keto**	1.432	27	0.4148	**Age**	F (1, 29) = 1.460	0.2367	**Aged**	1.568	29	0.239
	**Diet**	F (2, 27) = 5.718	0.0085	**Ad libitum Chow**	3.356	27	0.0071	**Feeding Paradigm**	F (1, 29) = 6.425	0.0169				
**Rarity**	**Interaction**	F (2, 27) = 0.1713	0.8435	**TRF Control**	0.6629	27	0.8845	**Interaction**	F (1, 29) = 0.3796	0.5426	**Young**	1.512	29	0.2625
	**Age**	F (1, 27) = 2.187	0.1508	**TRF Keto**	2.493	27	0.0563	**Age**	F (1, 29) = 1.495	0.2313	**Aged**	2.022	29	0.1023
	**Diet**	F (2, 27) = 3.253	0.0543	**Ad libitum Chow**	1.782	27	0.2363	**Feeding Paradigm**	F (1, 29) = 6.365	0.0174				

## Data Availability

Not applicable.

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
