# Peer review of "Influence of Aging, Macronutrient Composition and Time-Restricted Feeding on the Fischer344 x Brown Norway Rat Gut Microbiota"

_nutrients, 2022, doi:10.3390/nu14091758_

Round 1

Reviewer 1 Report

Interesting and well-designed study regarding the effects of manipulating daily feeding window, a new insight in nutrition. The paper is also quite innovative as it investigates the role of microbiome on health.

Here are some suggestions to make the paper even more complete;

- TRF is not the only type of intermittent fasting. You could expand the introduction briefly mention, at the beginning, other types of intermittent fasting. You can use for reference PMID: 26374764

- Regarding TRF protocols, do you investigate early or late TRF?  What are the differences of early or late TRF on gut bacteria?

- It would be useful to deepen the effects of TRF on mental health. You can use for reference PMID: 33356688

- It would be interesting to add more about TRF Keto studies in human

Author Response

- TRF is not the only type of intermittent fasting. You could expand the introduction briefly mention, at the beginning, other types of intermittent fasting. You can use for reference PMID: 26374764

  • We thank the authors for this suggestion. We have amended the introduction to include a broader description of TRF/IF and reference to the suggested manuscript (lines 43-45).

- Regarding TRF protocols, do you investigate early or late TRF?  What are the differences of early or late TRF on gut bacteria?

  • We have amended the methods section to include more details regarding the time of feeding (late TRF; lines 131-133). Unfortunately, we were not able to find any literature regarding differences in early or late TRF on gut microbiome composition.

- It would be useful to deepen the effects of TRF on mental health. You can use for reference PMID: 33356688

  • We thank the reviewer for this insightful comment and have amended the discussion section to include reference to this helpful review (lines 435-438).

- It would be interesting to add more about TRF Keto studies in human

  • While we agree with the reviewer that this would be interesting, there is limited literature regarding this combination in humans. However, we have attempted to include this in the introduction as best as possible (lines 100-104).

Reviewer 2 Report

The paper from Hernandez at al. adds exciting new insights in the beneficial effects of time-restricted feeding (TRF). Through the analysis of fecal microbiome  of FBN rats, a well-characterized model of aging, subjected to TRF with both Western-like standard diet and ketogenic diet, the authors were able to dissect the influence of diet and feeding paradigm  in the capability of TRF with both diets of improving age-related intestinal dysbiosis. Besides, through metagenomic analyses, the authors established functional differences between microbiomes of aged rats fed with different diets that  may drive the positive outcomes following dietary intervention.

I believe that the study is interesting, the study-design is adequate, and the paper is well written.

I only have one minor point: references at lines 374, 378, 401 and 402 in the Discussion section are cyted using the author’s name with the year, whereas all along the paper references are cited using a number within brackets.  Also these references have to be cited with their progressive number. References “Perry et al., 2016”, and “Si et al., 2018” are lacking from the Reference section.

Author Response

The paper from Hernandez at al. adds exciting new insights in the beneficial effects of time-restricted feeding (TRF). Through the analysis of fecal microbiome of FBN rats, a well-characterized model of aging, subjected to TRF with both Western-like standard diet and ketogenic diet, the authors were able to dissect the influence of diet and feeding paradigm  in the capability of TRF with both diets of improving age-related intestinal dysbiosis. Besides, through metagenomic analyses, the authors established functional differences between microbiomes of aged rats fed with different diets that may drive the positive outcomes following dietary intervention.

I believe that the study is interesting, the study-design is adequate, and the paper is well written.

I only have one minor point: references at lines 374, 378, 401 and 402 in the Discussion section are cyted using the author’s name with the year, whereas all along the paper references are cited using a number within brackets.  Also these references have to be cited with their progressive number. References “Perry et al., 2016”, and “Si et al., 2018” are lacking from the Reference section.

  • We appreciate the authors compliments regarding the manuscript. Moreover, thank you very much for pointing out the issues with the references. We aren’t sure why the reference manager did not catch this, but we have manually edited these references and appreciate the help.